# Rapid generation of homogenous tumor spheroid microtissues in a scaffold-free platform for high-throughput screening of a novel combination nanomedicine

**Hossein Abolhassani**[1◉], **Mohammad Zaer**[2◉], **Seyed Abbas Shojaosadati**[1]*, **Sameereh Hashemi-Najafabadi**[2]

**1** Faculty of Chemical Engineering, Biotechnology Group, Tarbiat Modares University, Tehran, Iran,
**2** Faculty of Chemical Engineering, Biomedical Engineering Department, Tarbiat Modares University, Tehran, Iran

◉ These authors contributed equally to this work.
* shoja_sa@modares.ac.ir

**Data Availability Statement:** All relevant data are within the manuscript and its Supporting Information files.

## Abstract

Combination nanomedicine is a potent strategy for cancer treatment. Exploiting different mechanisms of action, a novel triple drug delivery system of 5-fluorouracil, curcumin, and piperine co-loaded human serum albumin nanoparticles (5FU-CUR-PIP-HSA-NPs) was developed via the self-assembly method for suppressing breast tumor. Both hydrophobic and hydrophilic drugs were successfully encapsulated in the HSA NPs with a high drug loading efficiency (DLE) of 10%. Successful clinical translation of nanomedicines, however, is a challenging process requiring considerable preclinical *in vitro* and *in vivo* animal tests. The aim of this study was to develop a homemade preclinical 3D culture model in the standard 96-well plates in a cost and time-effective novel approach for the rapid generation of homogenous compact tumor spheroids for disease modeling, and anticancer therapeutic/nanomedicine screening. The knowledge of drug screening can be enhanced by employing such a model in a high-throughput manner. Accordingly, to validate the formulated drug delivery system and investigate the cellular uptake and cytotoxicity effect of the nanoformulation, 3D tumor spheroids were employed. The practicality of the nanomedicine system was substantiated in different tests. The *in vitro* uptake of the NPs into the tight 3D tumor spheroids was facilitated by the semi-spherical shape of the NPs with a proper size and surface charge. 5FU-CUR-PIP-HSA-NPs demonstrated high potency of migration inhibition as a part of successful anti-metastatic therapy as well. The remarkable differences in 2D and 3D cytotoxicities emphasize the importance of employing 3D tumor models as an intermediate step prior to *in vivo* animal experiments for drug/nanomedicine screening.

## 1. Introduction

Despite early detection, breast cancer is still considered a clinical concern [1]. Nanotherapeutics in many different formulations have been widely proposed for improving the efficacy of

**Funding:** This study was financially supported by Tarbiat Modares University (TMU) and Iran National Science Foundation (INSF). There was no additional external funding received for this study. The funders had no role in study design, data collection and analysis, decision to publish, or preparation of the manuscript.

**Competing interests:** The authors have declared that no competing interests exist.

cancer treatment by enhancing the bioavailability, tumor targeting, and stability of drugs while reducing their side effects [2]. However, in *vivo* tissue penetration of therapeutics/nanomedicines into solid tumors is influenced by the three-dimensional (3D) nature of the tumor microenvironment and drug resistance [3]. Alleviation of the transport barriers and multidrug resistance (MDR) could be achieved through the design of optimized drug delivery systems while exploiting practical *in vitro* tumor models for the nanomedicine screening [4].

In this work, a novel triple drug delivery system of 5-fluorouracil, curcumin, and piperine co-loaded human serum albumin nanoparticles (5FU-CUR-PIP-HSA-NPs) was synthesized via the self-assembly method to impose a synergistic effect on breast tumor tissue with different mechanisms of action. As a thymidylate synthase inhibitor, 5FU can interfere with the functions of DNA and RNA of cancer cells and inhibit cell growth [5]. Phytochemicals with biological effects against cancer cells can be used in combination with existing chemotherapeutics to enhance tumor therapy [6]. Accordingly, the effectiveness of 5FU as a chemotherapeutic can be improved through combinational treatment with CUR [5,7–9]. PIP, on the other hand, is capable of enhancing the bioavailability of drugs by suppressing the MDR effect through inhibiting P-glycoprotein (P-gp) activity [10,11]. Besides, CUR and PIP as natural bioactive products possess several biological and pharmacological properties such as anti-inflammatory, antioxidant, antiproliferative, anticancer, and antiangiogenic [10,12,13]. In addition to its biopharmaceutical properties, CUR, as a biocompatible probe for bio-imaging, emits strong fluorescence [14]. By encapsulating all these three drugs in a highly biocompatible nanocarrier i.e. HSA-NPs with proper physiochemical properties, the limitations associated with administration of the free drugs like poor water solubility, non-specificity, systemic toxicity, low plasma half-life, and MDR effect can be overcome and a tremendous effect of cancer theranostics can be achieved [5,11,15]. Using such a delivery system, both hydrophobic and hydrophilic drugs can be simultaneously encapsulated with a high drug loading efficiency (DLE).

Successful clinical translation of nanomedicines is a challenging process requiring considerable preclinical *in vitro* and *in vivo* animal studies [16]. In most research studies, drug delivery systems are screened on two-dimensional (2D) monolayer cell culture models followed directly by the *in vivo* animal studies [17]. Nevertheless, they cannot accurately resemble the tumor tissue and its microenvironment [18]. To address this, the tumor bioengineering approach has emerged and several preclinical models have been developed for better disease modeling and therapeutic screening [19,20]. The 3D platforms have the potential to bridge the gap between 2D cell cultures and *in vivo* animal models, enhance drug screening, and accelerate the clinical discovery and translation of findings leading to personal and precision medicine [21,22]. As a practical inexpensive 3D tumor model, spheroid microtissues have been extensively employed for drug screening due to their ease of formation and multicellular 3D structure [23]. However, The complexity of forming homogeneous tight spheroids, long-term culture need, low physiological relevancy, lack of reproducibility, difficulties in analysis and manipulation, and low throughput screening have been counted as the main disadvantages of the utilized tumor spheroid culture platforms so far [4,24,25].

To address these, herein, we established a preclinical scaffold-free 3D spheroid culture model in the standard 96-well flat plates by employing a homemade set-up. The novel aspect of our 3D platform includes the generation of a large quantity of uniform tight single spheroids in a cost and time-effective approach for drug screening with no necessary transfer step. Using this new in situ high throughput compatible 3D model system alongside 2D culture platforms, we investigated the efficacy of the formulated drug delivery system (5FU-CUR-PIP-HSA-NPs) to triple negative MDA-MB-231 breast cancer cells. The study provides proof-of-concept results for employing the 3D culture system to test the biological activities of the drug delivery system.

## 2. Materials and methods

### 2.1. Materials

HSA, dithiothreitol (DTT), 3-[4,5-dimethylthiazol-2-yl]-2,5-diphenyltetrazolium bromide (MTT), polyvinyl alcohol (PVA), pluronic F-127, fluorescein diacetate (FDA), and propidium iodide (PI) were purchased from Sigma-Aldrich (The United States). SYLGARD$^{®}$ 184 (Poly-dimethylsiloxane (PDMS)) was provided by Dow Corning (The United States). 5FU and CUR were obtained from Bio Basic Inc (Canada) and PIP was supplied from Alfa Aesar (The United States). BCA protein assay kit was purchased from Thermo Fisher Scientific (The United States). All other chemicals, reagents, and solvents were of the analytical grades.

### 2.2. Synthesis of 5FU-CUR-PIP-HSA-NPs

The triple drug delivery system of 5-fluorouracil, curcumin, and piperine co-loaded albumin nanoparticles (5FU-CUR-PIP-HSA-NPs) was synthesized using the self-assembly method as previously described by Gong et al. [26]. In our previous works, the self-assembly technique was employed as a practical technique for the preparation of drug-loaded-HSA-NPs [11,27,28]. Also, in a systematic study, the factors affecting the method were optimized [27]. While employing the literature, some preliminary tests were performed to prepare 5FU-CUR-PIP-HSA-NPs with the optimal conditions in hand for the self-assembly method from our previous studies [7,8,11,27]. Briefly, 25 mg of HSA was dissolved in acetate buffer at pH 3.5. 1 mg of 5FU was added to the solution and its temperature was adjusted to 37 ˚C. As a reducing agent, DTT was added to the solution to the final concentration of 5 mM. After 15 min, under a constant stirring rate of 550 rpm, 0.5 ml of hydrophobic drugs solution (ethanol as solvent) containing 2 mg of CUR and 0.5 mg of PIP was added to the solution at the rate of 1ml/min. After the addition of the drugs solution, turbidity was observed as an indication of the NPs formation. The suspension vial was immediately put in an ice bath to stop the synthesis process and subsequent agglomeration of the NPs. The resulting NPs were washed three times. Each cycle of purification was performed by centrifugation of the suspension (20,000×g, 15 min) followed by redispersion of the pellets in distilled water utilizing an ultrasonic bath for 5 min.

### 2.3. NPs characterization

**2.3.1. Particle size distribution, polydispersity index, and zeta potential measurement.** The particle size distribution and polydispersity index (PDI) of the NPs were measured by dynamic light scattering (DLS) using the NANO-flex® apparatus (Particle Metrix, Germany) at 25 ˚C. The ZETA-check (Particle Metrix, Germany) was utilized to determine the surface charge of the NPs at room temperature.

**2.3.2. Determination of yield, drug loading, and encapsulation efficiency.** The supernatant obtained from the first cycle of centrifugations after the NPs synthesis was assessed to determine the yield, drug encapsulation efficiency (DEE), and drug loading efficiency (DLE) of the NPs. The unassembled HSA content was obtained using the Pierce BCA protein assay kit. Yield (%) of the NPs was calculated based on Eq (1).

$$\text{Yield} = \frac{\text{Initial weight of HSA} - \text{Weight of HSA in supernatant}}{\text{Initial weight of HSA}} \times 100 \qquad (1)$$

The supernatant was also examined to determine the amount of non-encapsulated CUR and PIP after the NPs synthesis using a spectrophotometer at 427 nm and 343 nm, respectively. The amount of 5FU in the supernatant was measured utilizing a high-performance liquid chromatography (HPLC) apparatus (Young Lin HPLC System, South Korea). A mixture of

methanol/acetic acid (80:20, v/v) was used as the mobile phase and eluted isocratically through a reversed-phase column ($C_{18}$ MZ-Analytical column, 200 mm × 4.6 mm, 5 mm, ODS-3) with a flow rate of 1 ml/min at room temperature. 50 μl of diluted supernatant was loaded into the column and the detective wavelength of 270 nm was set to analyze the 5FU amount. The concentration range of 0.005 to 0.1 mg/ml was used to plot the linear calibration curve with a correlation coefficient of $R^2 = 0.99$. The drug content in the NPs was obtained by subtracting the amount of free drugs in the supernatant from the total amount of the drug initially used. DLE (%) and DEE (%) were calculated using Eqs (2) and (3), respectively.

$$\text{DLE (\%)} = \frac{\text{Weight of drugs in NPs formulation}}{\text{Weight of 5FU} - \text{CUR} - \text{PIP} - \text{HSA} - \text{NPs}} \times 100 \tag{2}$$

$$\text{DEE (\%)} = \frac{\text{Weight of drug in NPs formulation}}{\text{Weight of initial drug used}} \times 100 \tag{3}$$

**2.3.3. Differential scanning calorimetry (DSC) analysis.** The thermal behavior of HSA, 5FU, CUR, PIP, and freeze-dried powder of 5FU-CUR-PIP-HSA-NPs was investigated using the DSC instrument (Mettler Toledo, Switzerland). The samples were sealed in aluminum pans and analyzed over a temperature range from 30–350 ˚C at a rate of 15 ˚C/min under a nitrogen atmosphere.

**2.3.4. Morphology analysis of 5FU-CUR-PIP-HSA-NPs and tumor spheroids.** A field emission scanning electron microscope (FE-SEM) (Carl Zeiss, Germany) was employed to examine the structure of the NPs and 3D tumor spheroids. To evaluate the morphology of the NPs, 15 μl of 5FU-CUR-PIP-HSA-NPs was placed on an aluminum stub and dried under a vacuum condition. The dried sample was coated with a gold layer and observed with the instrument.

After 7 days post-seeding, a spheroid formed on the concave well in the 3D platform was washed with phosphate-buffered saline (PBS) twice and fixed with 2.5% (v/v) glutaraldehyde in deionized water at 4 ˚C for 90 min. Followed by twice washing, dehydration of the sample was performed by subjecting the spheroid to increasing gradients of ethanol solutions in deionized water (25%, 50%, 75%, 95%, and 100%) for 5 min. The sample was dried at room temperature, sputtered with a gold layer, and analyzed with the FE-SEM apparatus.

**2.3.5. *In vitro* drug release study.** Using the dialysis bag method, the *in vitro* drug release was studied in two different release buffers including PBS at pH 7.4 and acetate buffer (representative of the tumor microenvironment) at pH 5.5. To obtain the release patterns, 1 ml of free drug solutions (0.5 mg/ml) or 5FU-CUR-PIP-HSA-NPs suspension (containing 0.7 mg CUR, 0.38 mg 5FU, and 0.1 mg PIP) was placed in a dialysis bag (MW: 6–8 KDa), immersed into 20 ml of buffers, and assessed at 37˚C while stirring at 100 rpm for 48 h. 0.5% (w/v) sodium dodecyl sulfate (SDS) was added to the release buffers to maintain a sink condition and facilitate the release of the hydrophobic drugs [29]. At predetermined time intervals, samples (1 ml) were withdrawn and replaced by 1 ml of fresh medium. The amount of released drugs was determined and quantified using the standard curve of each drug.

## 2.4. 2D cell culture and 3D spheroid formation

The metastatic triple-negative breast cancer cell line, MDA-MB-231, was obtained from Pasture Institute, National Cell Bank of Iran. Cells were cultured in Dulbecco's Modified Eagle's Medium (DMEM) supplemented with 10% fetal bovine serum (FBS) and 1% Streptomycin/ Penicillin. The cultured cells were incubated with 5% $CO_2$ at 37˚C in a humidified incubator.

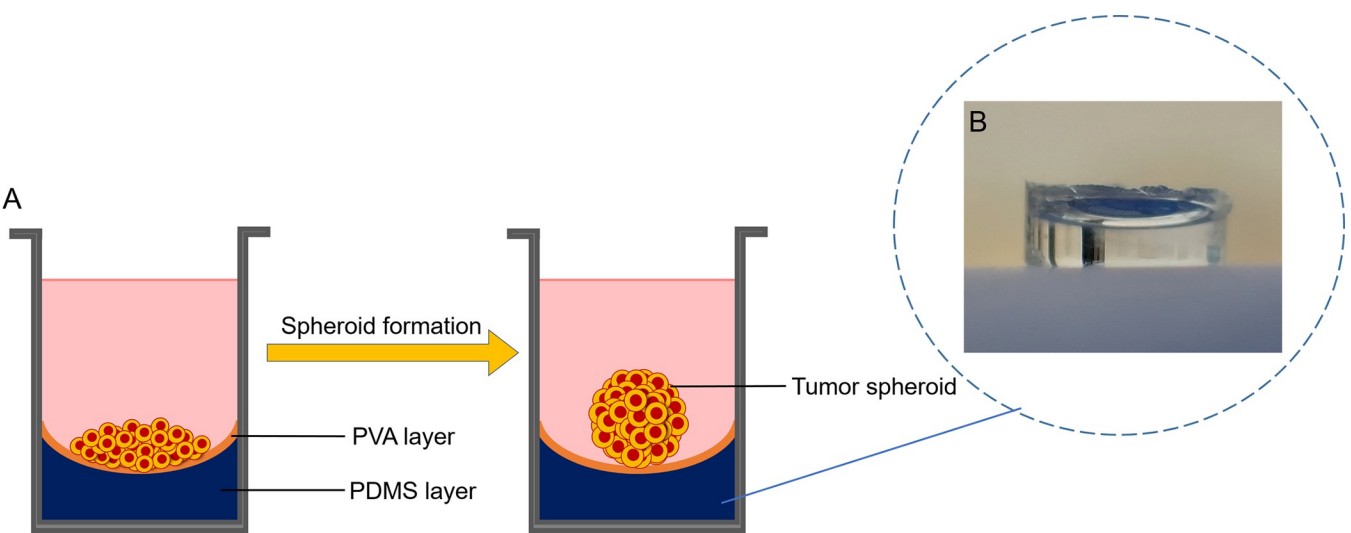

**Fig 1. 3D platform for spheroid culture.** (A) Spheroid culture process onto the non-adherent concave bed in a 96-well plate. (B) peeled off PDMS layer from a well.

2D monolayer culture of MDA-MB-231 cells was provided for different tests by seeding 20,000–30,000 cells/well into a 96-well plate. The bioassays were performed on 2D cell cultures after attachment to the plate surface over 24 h.

Rapid scaffold-free spheroid culture of MDA-MB-231 cells was performed in a cost-effective novel approach by employing the liquid overlay technique principles [30]. A treated concave-based bed was provided for cells in 96-well flat-bottom plates to form homogenous tumor spheroids (Fig 1A and 1B). Initial optimization tests were implemented to define the optimum values affecting the process (eg. PDMS volume, coating substrate, cell density, culture time). To this end, a pre-polymer of PDMS and its curing agent was prepared in a ratio of 10:1 w/w, stirred manually, and degassed in a vacuum desiccator. 50 µl of the mixture was poured into each well of the 96-well plate and allowed to settle and cast. The plate was placed in the incubator and the PDMS layer was cured at 37˚C for 5 h. Then, 50 µl of PVA solution (1% w/v in water) was poured into each well and incubated under a cell culture hood at room temperature for 24 h. The surface of the PDMS-cured 96-well plate was treated with PVA to prevent cell adhesion to the surface and facilitate tumor spheroid formation. The next day, the dried treated wells were washed with PBS and the plates were sterilized using UV for 30 min.

In order to produce large spheroids, high cell numbers have been used per well in 96-well plates [31–34]. The size of the spheroids can be manipulated by varying seeded cell numbers and culture periods [32]. MDA-MB-231 cells were reported to have a diameter and volume of 14.3 µm and 1530 µm$^3$, respectively [35]. Taking the ideal round shape and high tightness for tumor spheroids into account, the approximate number of cells needed for the generation of tumor spheroids with a desired size range was calculated using Eq (4).

$$V = \frac{1}{6}(\pi d^3) \tag{4}$$

Where D and V are the diameter and volume of the tumor spheroids, respectively. Performing some pretests under these assumptions, a cell number range of 20,000–40,000 was selected to obtain spheroids with a size range of 400–500 µm for different biological activity tests. Employing a high cell density of $10^6$ cells/ml, cells were seeded onto each non-adherent

round-bottomed well and were allowed to settle into the recesses. The medium then was added to each well to reach a volume of 200 μl. 50% of the culture medium was exchanged every other day during the tissue growth. Tumor spheroid microtissues were grown for 72 h to bear a compact structure before the biological activity tests. Spheroid formation was monitored over time using an Optika microscope (Italy) with 4X and 10X magnifications.

## 2.5. Live/dead staining of a tumor spheroid

A tumor spheroid was grown for 4 days in the 3D platform by seeding 30,000 cells in a well. The spheroid was stained for the assessment of cell viability using the FDA/PI solution. The spheroid was exposed to the staining solution containing 8 μg/ml of FDA and 20 μg/ml of PI for 30 min in dark and then washed with PBS twice before imaging by fluorescence microscopy.

## 2.6. Flow cytometric apoptosis analysis of tumor spheroid

Breast cancer MDA-MB-231 tumor spheroids were cultured for 4 days using the initial density of 30,000 cells per spheroid in the provided 3D platform in a 96-well plate. The spheroids were washed with PBS and trypsinized to obtain single cells. Dissociation of cells was almost complete within 15 to 20 min while pipetting every 5 min. For apoptosis analysis, cell suspensions of 4 dissociated spheroids were pooled, stained with Annexin V-FITC and propidium iodide (PI), and studied by a flow cytometer (BD FACSCalibur™, The United States). Quadrant statistics were applied on the dot plots with the percent of viable cells located in the lower-left quadrant (Annexin V⁻/PI⁻).

## 2.7. *In vitro* uptake study

*In vitro* uptake of the free drugs combination and the NPs formulation was evaluated throughout the monolayer cells and tumor spheroids. Monolayer cell culture and tumor spheroids were grown by seeding 20,000 cells per well in 2D and 3D platforms, respectively. The cells in 2D and 3D models were exposed to the native drugs combination (CUR/5FU/PIP: 10/5.4/1.5, and 50/26.8/7.4 μg/ml) or 5FU-CUR-PIP-HSA-NPs at the equivalent concentrations for 4 h and 24 h. Afterward, the medium was discarded and cells were washed with PBS twice. The distribution of free drugs combination and the NPs in cells was observed using an inverted fluorescence optical microscope (TE2000, Nikon, Yokohama, Japan) equipped with a digital camera (DS-Qi1MC, Nikon, Yokohama, Japan) with 10X magnification.

## 2.8. Anticancer activity test in 2D and 3D platforms

MTT assay was employed to evaluate the anticancer activity of the free drugs combination and the prepared drug delivery system (5FU-CUR-PIP-HSA-NPs) on MDA-MB-231 cells at different concentrations in 2D and 3D conditions. Monolayer and tumor spheroid of the cancer cells were generated in 96-well plates by seeding 30,000 cells per well in 2D and 3D platforms, respectively. 2D monolayer cells and 3D tumor microtissues were exposed to the treatments after 24 h and 96 h cultures, respectively. The media were discarded and exchanged by the culture media containing different CUR/5FU/PIP concentrations (between 10/5.4/1.5 and 50/26.8/7.4 μg/ml) of free drugs combination or the equivalent doses of drugs in 5FU-CUR-PIP-HSA-NPs. The treatment of cells was performed in the incubator for 48 h. After the incubation time, the test media were removed. 100 μl of MTT solution (0.5 mg/ml in PBS) was placed in each well and incubated with 5% $CO_2$ at 37˚C for 4 h. Afterward, the MTT solution in each well was replaced by 150 μl of dimethyl sulfoxide (DMSO) to dissolve the formazan

crystals produced by the activity of mitochondria of the viable cells. In order to practically dissolve the formazone crystals in the 3D platform, a shaker was employed at 150 rpm for 20 min. The absorbance of formazan was read at the wavelength of 570 nm using an ELISA plate reader. The cell viability (%) was determined by calculating the ratio of mean absorbance of treated wells (n = 4) relative to the control group. The wells containing tumor cells with no treatments were considered the control group in both 2D and 3D platforms.

## 2.9. Scratch wound healing assay

The ability of 5FU-CUR-PIP-HSA-NPs in modifying MDA-MB-231 cancer cell motility was evaluated by the scratch wound healing assay [36]. In order to provide the desired confluency, high cell numbers of MDA-MB-231 cancer cells can be used per well in 24-well plates for the scratch wound healing assay [37,38]. In this study, the metastatic breast tumor cells were seeded into a 24-well plate with an initial density of 140,000 cells/well. After the attachment and reaching the approximate 70% confluency, the cell monolayer was scratched by a 200 µl sterile pipette tip and the debris was washed with PBS. MDA-MB-231 cells were then incubated with the medium containing a low CUR/5FU/PIP concentration of 10/5.4/1.5 µg/ml in the NPs. The wells containing tumor cells with no treatments were considered the control group. The changes in cell migration were monitored over time by optical microscopy. Images were captured at t = 0, 12, and 24 h at the same position of each well.

## 2.10. Statistical analysis

The experiments were repeated at least three times and the data were reported as mean ± standard deviation (SD). GraphPad Prism 7.0 software was utilized for the statistical analysis of the results. One-way analysis of variance (ANOVA) with repeated measures was used for comparison within a group of normally distributed data. For multiple group comparisons, two-way ANOVA was applied. In order to analyze the significance of differences and correct for multiple comparisons, Tukey's post-hoc test was performed. The differences were considered statistically significant at a p-value $< 0.05$.

## 3. Results and discussion

### 3.1. 5FU-CUR-PIP-HSA-NPs preparation and characterization

To exploit different mechanisms of action in breast tumor treatment, the triple drug delivery system of 5FU, CUR, and PIP was synthesized using the self-assembly method. Both hydrophobic and hydrophilic drugs were successfully encapsulated in the HSA NPs as a single drug delivery system. In the self-assembly process, the intermolecular disulfide bonds of albumin molecules are cleaved and hydrophobic regions are uncovered by using DTT as a reducing agent. The added hydrophobic drugs to the medium, make bridges between HSA molecules through hydrophobic interactions and result in semi core-shell self-assembled NPs [26]. CUR has a strong binding affinity to HSA at the hydrophobic cavities of the protein [39]. PIP, on the other hand, binds with HSA majorly by hydrophobic interactions along with a few hydrophilic interactions [40]. The decisive role of hydrophobic interactions between lipophilic drugs and albumin molecules in creating bridges, self-assembly of NPs, and drug loading have been indicated in several studies [11,26,28,41]. During the synthesis process, 5FU, the water-soluble drug that existed in the albumin solution could be entrapped in the NPs as well. Hydrophobic interactions and hydrogen bonding were reported to stabilize the 5-FU interaction with HSA [42].

The NPs were synthesized with a yield of 85.1% ± 1.3% and demonstrated a high negative surface charge of −25.4 ± 3.6 mV and mean particle size of 158.3 ± 5.6 nm with a PDI of 0.15. The high negative charge of the NPs could guarantee the stability of the drug delivery vehicle. DEE of 75.38% ± 1.4%, 70.22% ± 3.5%, and 41.5% ± 2.7% were calculated for 5FU, CUR, and PIP, respectively. Considering the loading of all drugs, a significant DLE of 10% was achieved for this formulation.

### 3.2. Morphology of NPs

The morphology of NPs, obtained by the FE-SEM analysis, indicated a semi-spherical shape for the NPs (Fig 2A). The high negative charge of the NPs could be a determining factor in inhibiting the NPs from aggregation. The results were in accordance with the determined mean particle size (158.3 ± 5.6 nm) in the DLS analysis (Fig 2B). The diameter of the NPs was less than 200 nm. NPs smaller than 200 nm were reported to have low uptake by opsonization, long circulation time, and enhanced accumulation in tumor tissue *in vivo* by the passive

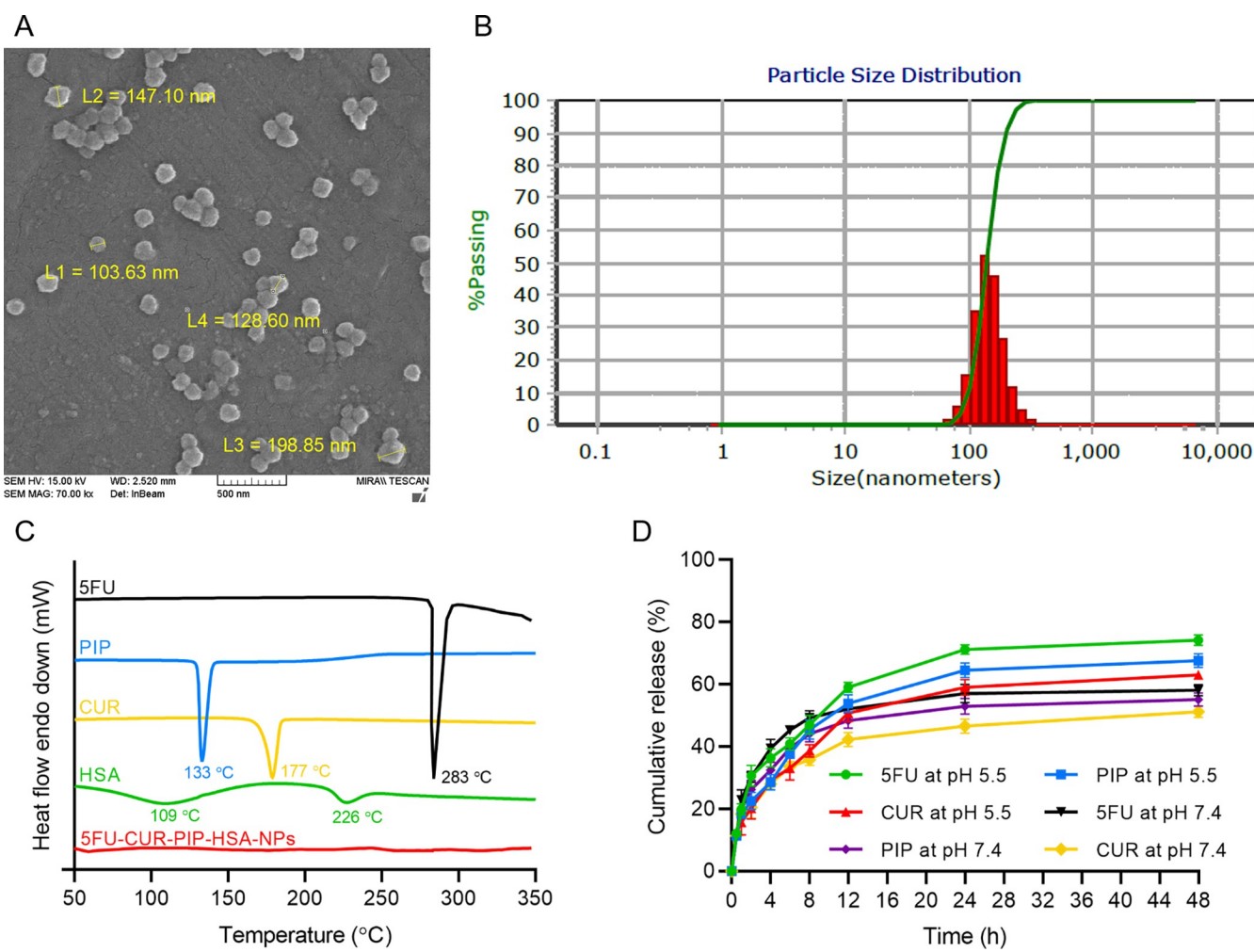

**Fig 2. 5FU-CUR-PIP-HSA-NPs characterization.** (A) SEM image and (B) particle size distribution of 5FU-CUR-PIP-HSA-NPs. (C) DSC thermograms of 5FU, PIP, CUR, HSA, and 5FU-CUR-PIP-HSA-NPs. (D) *In vitro* release patterns of the three drugs from 5FU-CUR-PIP-HSA-NPs in different buffers at 37˚C. Each point represents the average of three independent experiments, and error bars show the SD.

targeting through the enhanced permeability and retention (EPR) effect. The EPR effect arising from the disarrangement of endothelial cells in tumor vasculature and an impaired lymphatic system facilitates the permeation and accumulation of the NPs with a proper size into the tumor region [43].

### 3.3. DSC characterization

DSC thermoanalytical technique was employed to analyze the thermal behavior of the prepared NPs and demonstrate the state of the drugs in the nanoformulation. The thermocurves of 5FU, PIP, CUR, HSA, and 5FU-CUR-PIP-HSA-NPs were illustrated in Fig 2C. HSA showed endothermic peaks at 109˚C and 226˚C. 5FU, PIP, and CUR were characterized in the DSC diagram by the existence of endothermic peaks at 283˚C 133˚C, and 177˚C, respectively. The peaks are attributed to the melting points of the drugs and indicate their crystalline native form. The corresponding endothermic peaks of all the three drugs were not detected in the DSC curve of 5FU-CUR-PIP-HSA-NPs suggesting the conversion of the crystalline structure of the drugs to an amorphous state upon molecular binding with HSA molecules or physical entrapment into the NPs in the synthesis process. The amorphous state of the drugs within the NPs is preferred and could result in better dissolution, absorption, and bioavailability [44]. The absence of the characteristic peaks of the drugs confirms the existence of all the drugs (both hydrophobic and hydrophilic) within the NPs and implies the stability of the formulated drug delivery system.

### 3.4. *In vitro* release

The *in vitro* release of drugs from 5FU-CUR-PIP-HSA-NPs was tested in PBS and acetate buffer at 37˚C over 48 h. As presented in Fig 2D, an initial burst release of approximately 20% of total drug content was followed by a sustained release from the nanoformulation. At pH 7.4, after 48 h, the approximate cumulative release of 5FU, CUR, and PIP from 5FU-CUR-PIP-HSA-NPs was 58%, 51%, and 55%, respectively. While at pH 5.5, 74%, 63%, and 67% releases were recorded for 5FU, CUR, and PIP, respectively. On the other hand, the entire content of free drugs can be released within hours [11,27].

The release and diffusion of drug molecules, non-encapsulated or located near the inner surface of the NPs, could be responsible for the initial burst release profiles observed in the first hour. Hydrophobic interactions and hydrogen bonding between drugs and HSA molecules could control the release of the drugs from the NPs into the media. The diffusion and release of encapsulated drugs from the core-shell structure of the NPs could be accounted for the sustained release profile after the burst release.

As there are no electrostatic interactions between the drugs and HSA molecules, the release patterns of drugs are not substantially affected by lowering pH. However, a slightly enhanced drug release (approximately 15%) was observed at pH 5.5. Moreover, compared with CUR and PIP, 5FU showed higher cumulative drug release at pH 5.5 after 48 h ($p$-value $< 0.05$). 5FU with a smaller size and molecular weight has weaker or no interactions with HSA molecules in the nanoformulation. As a result, it can diffuse out easier from the NPs leading to a higher cumulative drug release.

### 3.5. Formation and characterization of multicellular tumor spheroids

Spheroids are 3D self-assembled cell aggregates with a spherical shape that can recapitulate chemical gradients (e.g. oxygen, nutrients, signaling, or catabolites) [4]. Tumor spheroid platforms are suitable models to obtain complementary information on nanomedicine efficacy and predict the *in vivo* potency of therapeutics. They have the potential to bridge the gap

between 2D *in vitro* cultures and *in vivo* animal models and enhance drug screening. The complex 3D structure of tumor spheroids can impact the penetration and distribution of therapeutics/nanomedicines; such domination and barriers are not represented in 2D cultures [45]. As practical models, 3D tumor spheroids have gained interest over the traditional 2D culture platforms and have been employed as suitable tools to study different physicochemical properties and the efficacy of drug delivery systems [4].

Several methods such as hanging drop, liquid overlay, magnetic levitation, micropatterned plates, spinner flasks, matrix encapsulation, scaffolds/hydrogels, and microfluidic devices have been employed for spheroid formation. Each technique has its distinct advantages and disadvantages depending on the application [4,24,46]. The tumor spheroids produced by different methods vary in size, culture time, or mechanical accessibility and could have different cellular organization and drug sensitivity due to their morphology characteristics like spheroid volume and shape [47–49]. In this regard, the ultra-low attachment microplates and polyethylene glycol dimethacrylate (PEGDMA) hydrogel microwell arrays have been reported to produce reproducible uniform size-controlled and dense 3D multicellular cancer spheroids compatible with cancer drug research and high-throughput screening [50]. In the ultra-low attachment plates, agar/agarose and poly(2-hydroxyethyl methacrylate) (pHEMA) are mainly used to coat the plates for spheroid culture [4]. The transferring step need for subsequent bioassays and the short storage time of the coated plates have been considered the main disadvantage of using agar/agarose in the liquid overlay technique [30,31], while long-term culture may be needed for spheroids to form and become dense [51,52]. In addition, when using flat non adherent surfaces, several irregular cellular aggregates can be formed limiting the high-throughput screening applications [30]. Besides, pHEMA is relatively expensive, and preparing the pHEMA-coated plates is laborious [30]. Moreover, compared with homemade coated plates, commercially available low-adhesion 96-well plates are relatively expensive [53].

To address these challenges, a particular treated concave-based bed in a 96-well plate (high throughput compatible) with a U-shape design was established in this study for rapid one spheroid formation in each well. PDMS, as a biocompatible cost-effective polymer with several excellent material properties for biomedical applications provides a chemically inert surface with low interfacial free energy, good thermal stability, and optical transparency. The surface properties of PDMS are also relatively easy to modify [54]. To provide the concave bed, PDMS was added into each well of a 96-well flat-bottom plate and the wells were coated with PVA providing an environment that inhibits cell attachment, protein absorption, and enzyme activation. Cell aggregation on a surface is controlled by two competing forces, cell-cell and cell-substratum interactions; when cell-substratum adhesivity is decreased, intercellular aggregation is promoted [32]. Upon seeding, cells were pelleted at the bottom of the wells owing to the provided concave bed of wells ensuring the initiation of single spheroid formation in each well. Spheroid cultures of uniform sizes and growth characteristics are critical factors that considerably affect the accurate quantification of biological or biochemical endpoints in drug/nanomedicine screening [31]. Compared with conventional techniques like hanging drop, this platform can rapidly generate reproducible homogenous large tumor microtissues with high compactness paving the way for a variety of high throughput screening assays and readouts with no transferring steps.

The tightness of the produced tumor spheroids with this method was intensified over time in 7 days with slight size variation after day one (Fig 3A). The formation of spheroids can be mediated mainly by the functional activity of integrin ß1 molecules in MDA-MB-231 cells [55,56]. The generation of MCF-7 tumor spheroids and co-culture spheroids (MDA-MB-231/MCF-7 cells and fibroblasts) was also feasible using this platform.

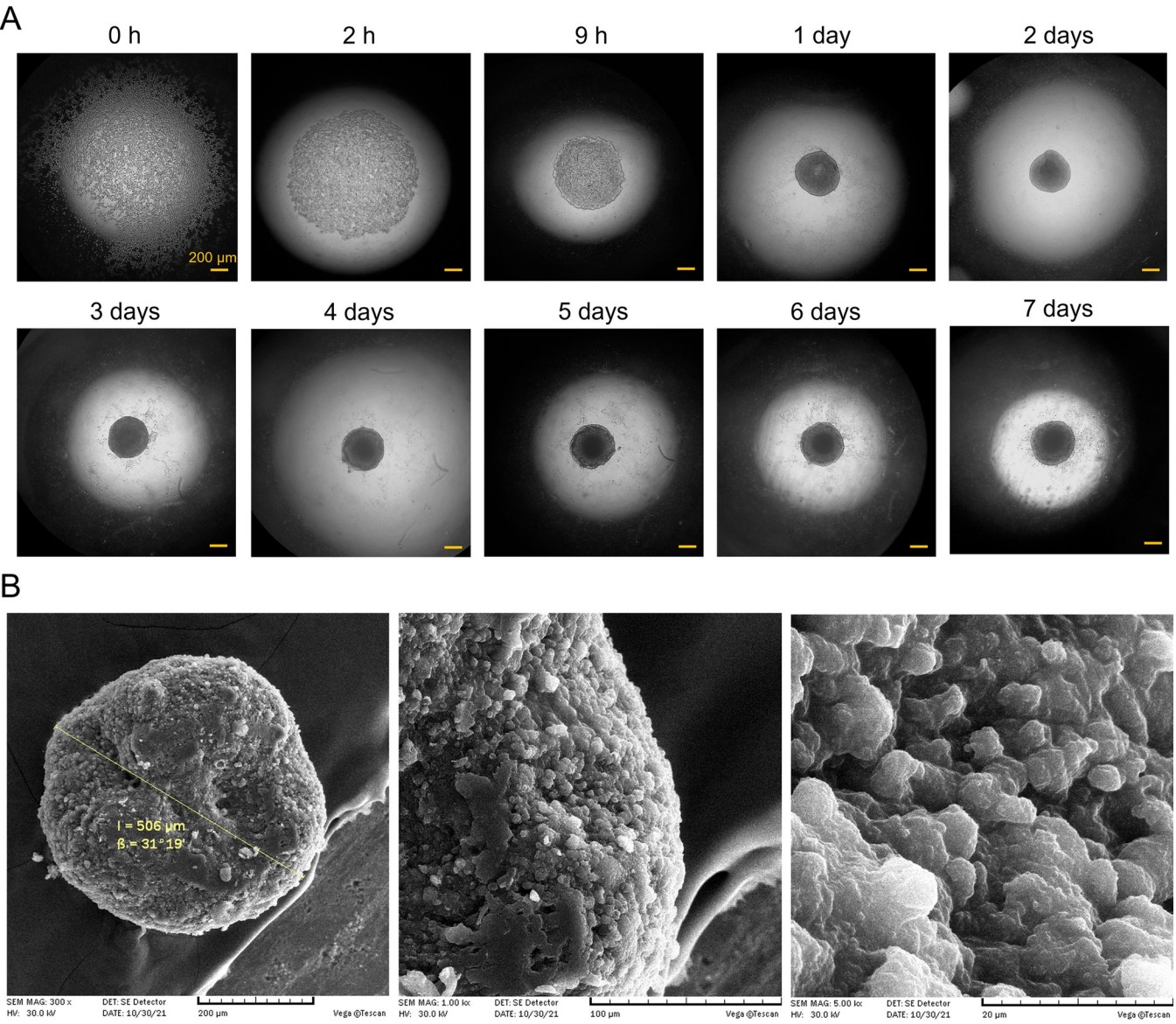

**Fig 3. Formation of tumor spheroids.** (A) Sequential images of spheroid culture with the initial cell density of 30,000 cells/well. The images were taken with 4X magnification by an optical microscope. Scale bars = 200 μm. (B) SEM images of the generated MDA-MB-231spheroid with the initial density of 40,000 cells per spheroid on day 7 with different magnifications.

Besides PVA, BSA and pluronic F-127 were utilized for PDMS coating and spheroid formation. Different substrates such as bovine serum albumin (BSA), PVA, and pluronic F-127 have been proposed for surface treatment for 3D cell culture in the literature [57]. The PDMS layer in wells was treated with 3% (w/v) BSA (undenatured or denatured), 1% (w/v) F-127, or 1% (w/v) PVA in water for spheroid culture. The results showed that F-127 and PVA treatments can yield better tumor spheroids in the 96-well plate (S1 Fig).

The size of tumor microtissues formed on the nonadherent surface can be adjusted based on experimental needs by varying the cell density and culture time. The technique introduced in this study offers several significant advantages over commonly employed methods (e.g. hanging drop, magnetic levitation, and spinner flasks) for spheroid formation (S1 Table). The

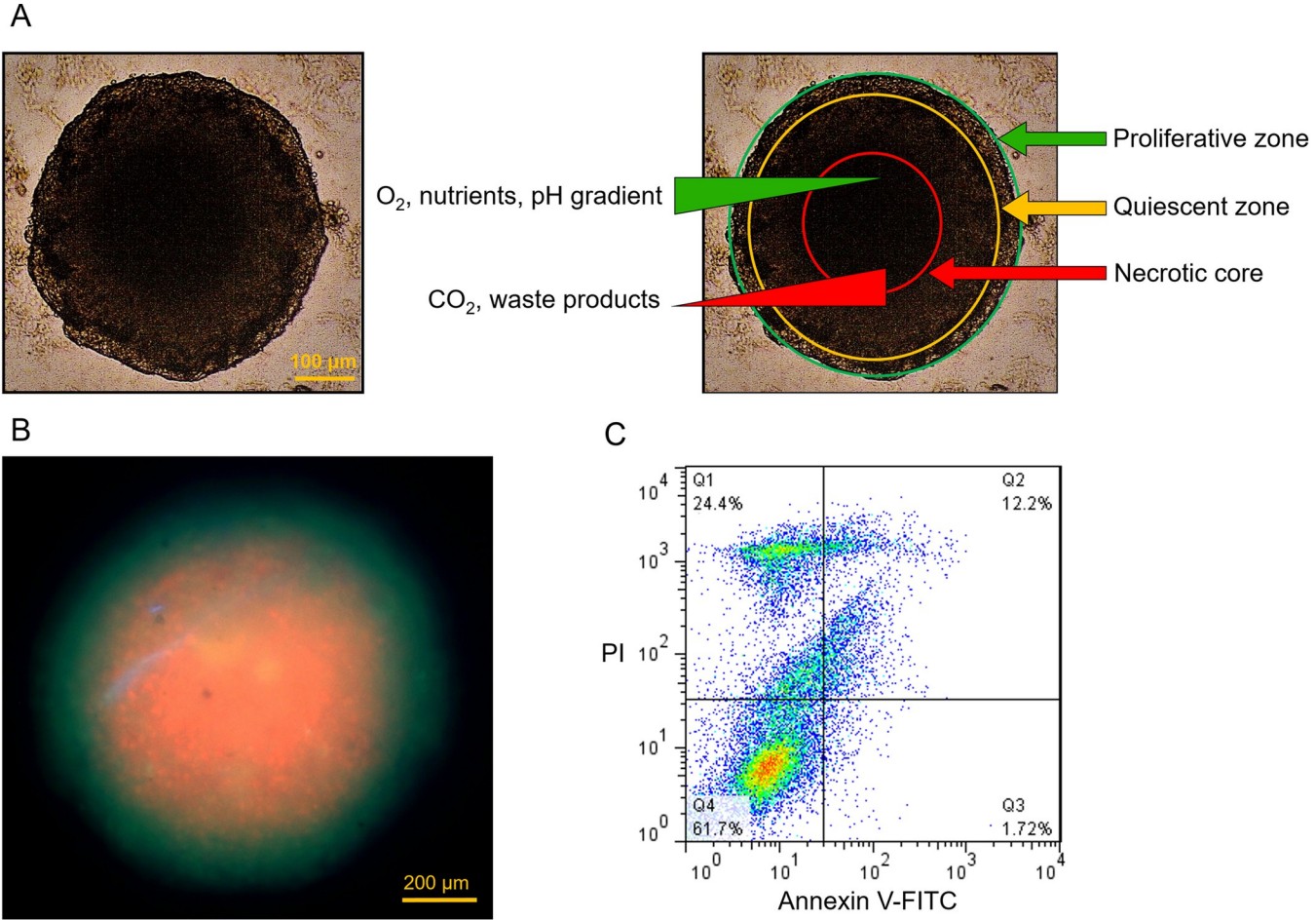

**Fig 4. Tumor spheroid characterization.** (A) Brightfield image of the tumor spheroid and the same image with pathophysiological concentric zones. Scale bar = 100 μm (B) Fluorescent image of live/dead assay on a tumor spheroid using FDA (green) and PI (red)). Scale bar = 200 μm. (C) Flow cytometric annexin V/PI analysis of MDA-MB-231 spheroids after 4 days. Quadrant assignment: Lower left = viable cells; lower right = early apoptotic cells; upper right = late apoptotic cells; upper left = necrotic cells.

spheroids in the platform can be harvested intact from the wells for downstream assays like SEM analysis and cryosectioning for histological examination.

The SEM image of the 3D tumor confirmed the spherical structure that was previously indicated by optical microscopy and showed the high compactness of the tumor tissue (Fig 3B). Such round-shaped spheroids produced in this model can benefit mathematical modeling too [30]. The establishment of uniformed size large spheroids (0.3–1 mm), in a clinically biorelevant tissue model, is technically challenging. However, the size of 506 μm was attainable for the cultured spheroid in this platform which is highly practical in recapitulating the *in vivo* tumor microenvironment. 300–500 μm was reported to be the optimum diameter size for tumor spheroids to practically mimic the *in vivo* conditions in terms of hypoxia and proliferation gradients [4]. No oxygen limitation is reported to occur in spheroids up to 100 μm in diameter [58].

To be a biorelevant model of *in vivo* avascular tumor tissue, spheroids need to possess various characteristics. The spheroids formed in this platform had proper size alongside mimicking the internal cellular layers and pathophysiology conditions that are present in tumor tissue *in vivo*. The representative three concentric zones of proliferative, quiescent, and necrotic regions can be characterized in the tumor spheroid formed in this platform (Fig 4A). The

existence of the necrotic core was substantiated by the live/dead staining of a tumor spheroid (Fig 4B). The platform supports the sustained growth of tumor spheroids with the capability of oxygen, nutrients, signaling, and metabolic gradient generation within large spheroids.

Similar to *in vivo* tumor tissue, tight organization of the cells was observed in the morphology images of the dense tumor spheroid (Fig 3B). The high tightness of the tumor structure can have a crucial impact on the development of a biochemical spheroid microenvironment recapitulating the avascular tumor tissue with a high percentage of dead cells in the inner core [55]. In a flow cytometry test, Ivascu and Kubbies represented the time kinetics of cell viability in tight-packed breast cancer spheroids and showed a continuous decrease in cell viability in spheroids over 6 days. Approximately, 80% and 60% viabilities were reported for MDA-MB-231 spheroids with an initial culture number of 10,000 cells per spheroid on days 1 and 4, respectively. Despite a decrease in the percentage of viable cells, the total number of viable cells was higher at the end of the culture period [49]. Similar to the reported value, 61.7% viability was detected in the flow cytometric annexin V/PI analysis of MDA-MB-231 spheroids after a 4-day culture period with an initial culture number of 30,000 cells per spheroids (Fig 4C) in this study. The low viability in the spheroid microtissue can be explained by the primary existence of the significant bulk of necrotic/apoptotic core in the spheroids. The 3D microenvironment of the tumor spheroid comprised of tight junctions creates a diffusion barrier for oxygen and nutrients in the inner layers of the microtissue [49].

### 3.6. *In vitro* uptake

Tumor microenvironment and drug penetration barriers in 3D solid tumors have been proven to be nearly as impactful as the MDR phenotype in drug resistance [59]. To enhance drug efficacy and distribution of therapeutics, nanoformulation and targeting approaches can be exploited. The potential of 5FU-CUR-PIP-HSA-NPs as a novel formulation of theranostic nanomedicine was evaluated through its uptake into MDA-MB-231 monolayer cancer cells and 3D tumor spheroids. Taking the advantage of strong intrinsic CUR fluorescence emission [14], the distribution of free drugs combination and the drugs delivered by the NPs was examined using fluorescence microscopy.

Intense CUR fluorescence was observed in the 2D monolayer platform for both the free drugs combination and NPs formulation indicating high cellular uptake in the cancer cells (Fig 5A and 5B). The *in vitro* uptake in tumor spheroids was demonstrated to be both time and concentration-dependent (Fig 5C and 5D). After 24 h incubation, the NPs achieved a homogenous radial distribution in tumor spheroids. Higher uptake and distribution were also detected for the NPs by increasing the drugs concentration.

By incubating the tumor spheroids with the free drugs combination and the NPs formulation, it was observed that free drugs were mostly confined to the periphery of the tumor spheroids which is in accordance with the limited penetration of free drugs observed in spheroids in the literature [52,60,61] indicating the persistence of binding barriers and decreased cellular interaction in produced tumor spheroids in our platform. On the other hand, deeper penetration was observed for the NPs formulation into spheroids (even into their necrotic core). Proper size, zeta potential, and drug combination of the prepared drug delivery system in albumin NPs could be responsible for the facilitated effective deep penetration into the solid tumors in good agreement with previous studies [4,62–64]. Negatively charged NPs have been demonstrated to be capable of penetrating deeply into tumor spheroids [4]. Diffusion of the nanoformulation into spheroids could also be mediated in the presence of P-gp inhibitors like PIP and CUR [65]. Altogether, the high potential of 5FU-CUR-PIP-HSA-NPs in bypassing

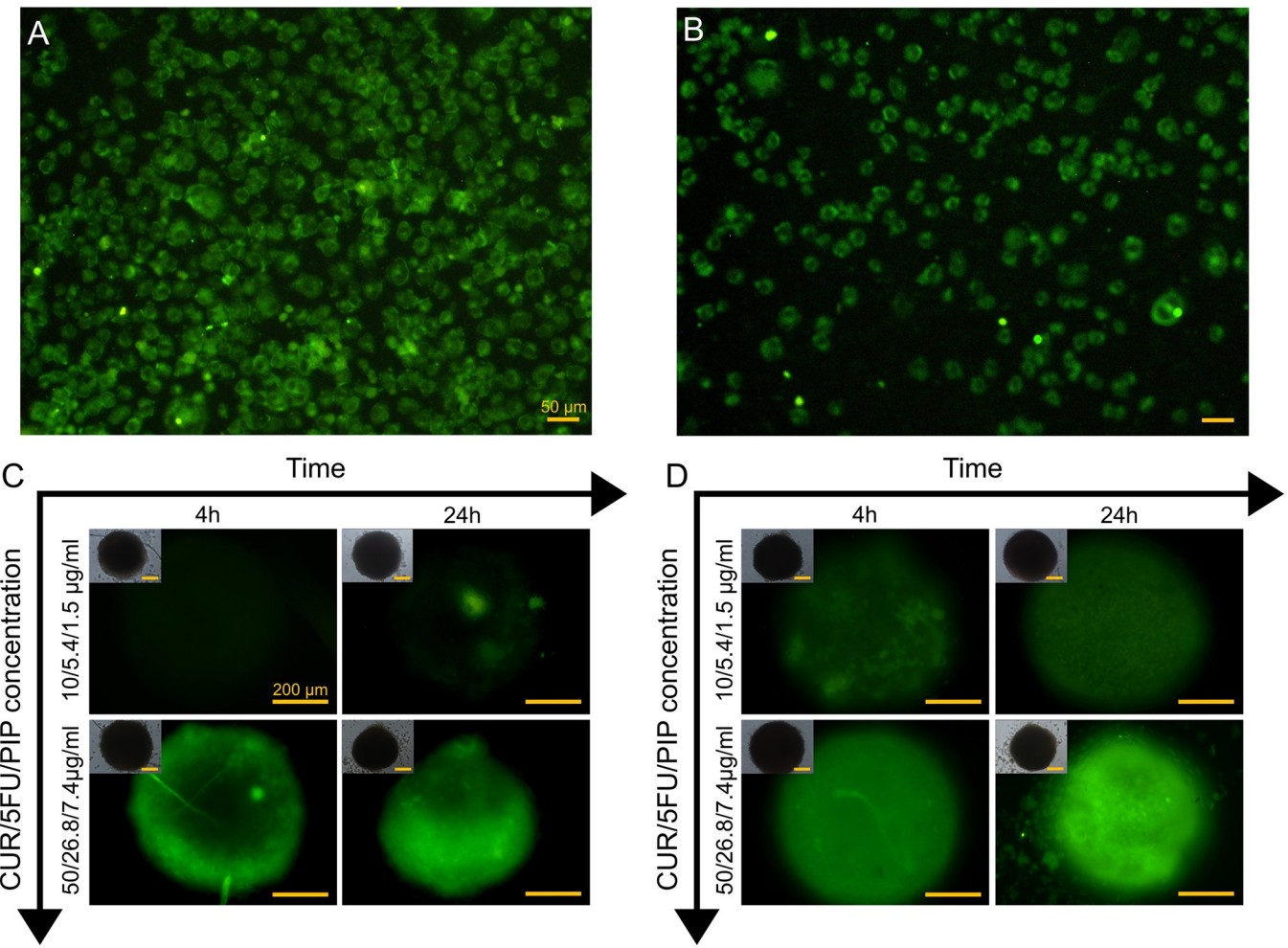

**Fig 5. Penetration of therapeutics/nanomedicine (5FU-CUR-PIP-HSA-NPs) in MDA-MB-231 cancer cells in 2D and 3D conditions.** *In vitro* uptake of (A) free drugs combination and (B) 5FU-CUR-PIP-HSA-NPs in 2D monolayer cancer cells at the CUR/5FU/PIP concentration of 50/26.8/7.4 μg/ml after 4 h. Scale bars = 50 μm. *In vitro* uptake of (C) free drugs combination and (D) 5FU-CUR-PIP-HSA-NPs with different concentrations and incubation times in 3D tumor spheroids. The green fluorescence represents CUR. Scale bars = 200 μm.

factors like physical penetration barriers effect, thereby effective tumor penetration and cancer theranostics was substantiated.

### 3.7. Anticancer activity in 2D and 3D culture platforms

The *in vitro* anticancer activity of free drugs combination and 5FU-CUR-PIP-HSA-NPs at equivalent concentrations was evaluated by the MTT assay on MDA-MB-231 monolayer cancer cells and 3D tumor spheroid microtissues after the 48 h treatment. No significant cytotoxicity was detected for bare HSA-NPs. Dose-dependent cytotoxicity was observed for both the free drugs combination and NPs formulation in 2D and 3D cultures (Fig 6A). The cytotoxicity induced by the free drugs combination was higher compared with its nanoformulation form in both 2D and 3D culture platforms. Similar results were observed in the study by Kim et al. [66]. In the NPs, the cargoes need time and driving force to be released inside the cells. This delay of action was reflected in their cell viability values. Nevertheless, the development of a triple nanoformulation with high stability can overcome the limitation of free drugs and unlock

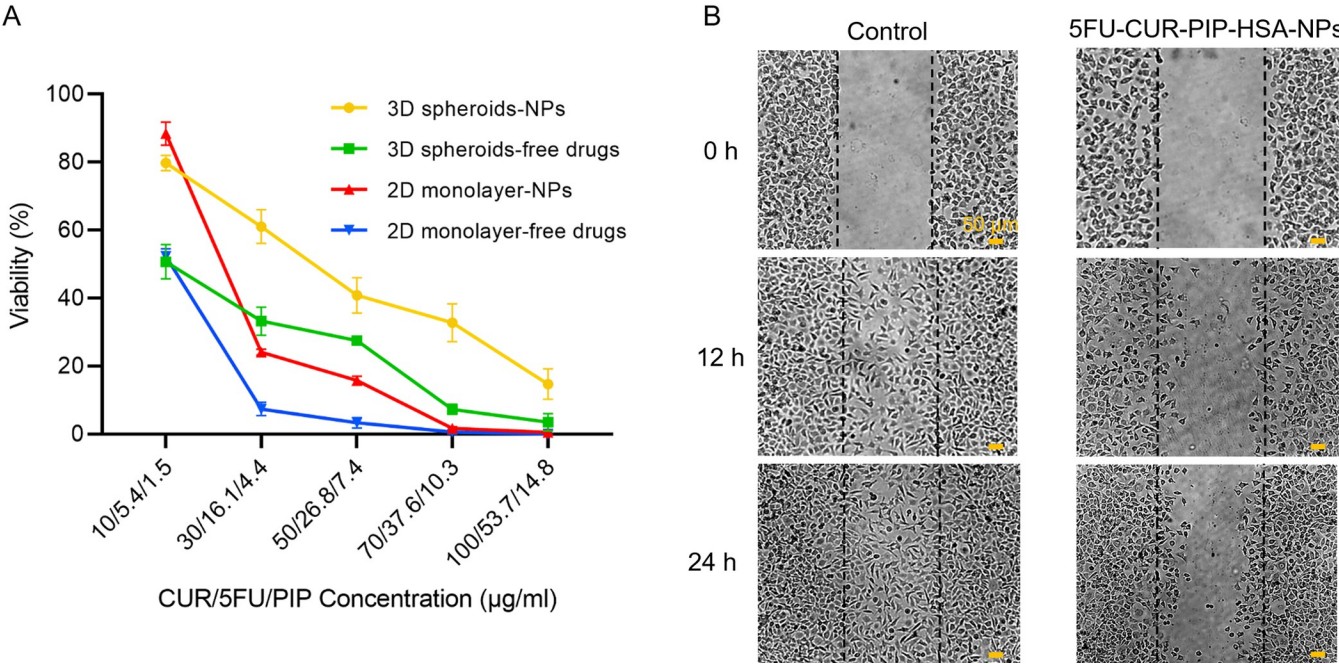

**Fig 6. Cytotoxicity and anti-metastatic activity of 5FU-CUR-PIP-HSA-NPs on MDA-MB-231 cancer cells.** (A) Anticancer activity of free drugs combination and 5FU-CUR-PIP-HSA-NPs on MDA-MB-231 monolayer (2D) and spheroids (3D) after the 48 h treatment. Each represented data point is calculated as the mean ±SD of four experiments. (B) Migration inhibition of metastatic MDA-MB-231 cells by wound healing assay. After the scratch creation, cells were treated with 5FU-CUR-PIP-HSA-NPs. After 24 h, the wound's gap became completely covered in the control group, meanwhile, the migration of metastatic cells was suspended using a low concentration of the NPs. Scale bars = 50 μm.

the full potential of their anticancer activity. The co-delivery approach provides the advantage of synchronizing the pharmacokinetics of different therapeutics, co-localizing the three drugs into the same tumor region, and maintaining a predetermined synergistic drug ratio leading to an enhanced efficacy [67]. Combinational treatment of 5FU, CUR, and PIP can have a synergistic effect on tumor tissue due to the different mechanisms of action of drugs [67]. The enhanced penetration and combinational therapy effect, previously observed in the *in vitro* uptake test were practically translated into the tumor spheroid cytotoxicity after the 48 h treatment (Fig 6A).

At the CUR/5FU/PIP concentration of 50/26.8/7.4 μg/ml, the free drugs combination and the NPs passed the half maximal inhibitory concentration ($IC_{50}$) in both 2D and 3D culture platforms. Strikingly up to 37% viability difference was observed between 2D and 3D cultures. At the CUR/5FU/PIP concentration of 70/37.6/10.3 μg/ml, the nanoformulation was capable of inducing 67.2% suppression in MDA-MB-231 spheroid cells; at the same concentration, the cancer cells were completely suppressed in the 2D monolayer culture. Compared with 2D culture platforms, the cells grown in 3D tumor structures possess different cell morphology, density, microenvironment, metabolism, and gene/protein expression which contribute to tight junctions, diffusion barriers, and drug resistance [18,68]. Dissimilar to the reported data [69,70], even at high concentrations, no destruction in the structure or reduction in size was observed in the tumor spheroids after the treatments.

Overall, the tumor spheroids were found to be considerably more resistant to the treatments than 2D monolayer cultures. Regardless of any physical barriers to drug penetration, there is a bulk of cells within the tumor spheroids more resistant to treatments than monolayer cultured cells. Hence, the higher cell viability of spheroids is expected to be a result of both

limited transport and intrinsic drug resistance associated with the 3D tumor microenvironment [71]. The remarkable differences in 2D and 3D cytotoxicities emphasize the importance of employing 3D tumor models prior to *in vivo* animal experiments for drug/nanomedicine screening.

## 3.8. Migration-cell wound closure assay

Breast tumor mainly shows signs of metastatic relapse in tissues like bone, lungs, liver, and brain [72]. Invasion of tumor cells is an initial step in the metastatic cascade. Besides suppressing tumor growth and killing cancer cells, another therapeutic approach is to inhibit the invasion and metastasis of tumor cells [73]. Wound-healing assay, as a standard and commonly used technique, was employed for the investigation of the cell migration in presence of the NPs formulation. Fig 6B represents the results of the wound healing assay. The control group demonstrated a dynamic migration toward the scratch center and the wound's gap became completely covered through the cells' migration after 24 h. On the other hand, by employing the low CUR/5FU/PIP concentration of 10/5.4/1.5 μg/ml for the NPs, the motility of the MDA-MB-231 cells was significantly suspended and the wound closure was not complete after 24 h treatment. A subtoxic concentration (CUR/5FU/PIP concentration of 10/5.4/1.5 μg/ml), lower than the $IC_{50}$ of the NPs (CUR/5FU/PIP concentration of 20.6/11.1/3 μg/ml), was used to minimize its cell-killing effect [37]. As shown in Fig 6A at low concentrations, there are significant differences between the anticancer activities of the free drugs combination and the NPs formulation. Consequently, since the free drugs combination group causes a high cell-killing effect, it cannot provide a reliable comparison for the NPs formulation in the wound healing assay. The results demonstrated that migration of the MDA-MB-231 breast cancer cell line with high migratory persistence can be intercepted by employing the formulated NPs in a time-dependent manner.

## 4. Conclusion

The potential of the triple drug delivery system (5FU-CUR-PIP-HSA-NPs) with a high drug loading and robust synergistic effects in cellular internalization, deep penetration in tumor spheroids, and cancer theranostics was substantiated in this study using both standard 2D cell culture and 3D tumor microtissue platforms. As a practical 3D model, a novel scalable scaffold-free production of controllable tumor spheroids platform was established in this study by utilizing a homemade set-up for disease modeling and anticancer drug/nanomedicine screening. The developed 3D platform mirrors at least some characteristics of the *in vivo* avascular tumor tissues and can serve as an intermediate step between monolayer cultures and *in vivo* models while the platform can be boosted by integrating other tumor microenvironment components like ECM and stromal cells into it. The high throughput screening approach can be applied to the employed method of spheroid formation in a 96-well plate with the conventional instrumentation for various bioassays and standardization and automated readout and imaging.

Exploiting such a 3D model, the efficacy of the triple drug delivery system was demonstrated in different tests. Relative to the cell monolayers, higher concentrations of drugs were needed to be employed for both free drugs combination and NPs formulation to induce cytotoxicity in spheroids due to the limited transport of the drugs/NPs into the tight structure of the microtissues and their intrinsic drug resistance. The highlighted efficacy of the triple nanoformulation in tumor penetration, anticancer activity, and migration inhibition suggests its success in more challenging *in vivo* animal models. Together, our results show that the assessment of nanomedicines by 3D tumor spheroid models as an intermediate step provides

valuable information regarding the efficacy of the formulations prior to *in vivo* animal tests, that otherwise cannot be obtained by 2D culture platforms. Translation of the employed 3D platform to wider applications (co-culture spheroids and organoid generation) for advanced drug screening is in development.

## Supporting information

**S1 Fig. Surface treatment for spheroid culture.** PDMS treatment with (A) Untreated, (B) 3% (w/v) BSA treated, (C) 3% (w/v) denatured BSA treated, (D) 1% (w/v) F-127 treated, (E) 1% (w/v) PVA treated. The initial cell density was 30,000 cells/well. The images were taken with 4X magnification by an optical microscope.
(TIF)

**S1 Table. Technological aspects of the presented 3D platform for tumor spheroid formation.**
(DOCX)

**S1 File. Minimal data set.**
(RAR)

## Acknowledgments

The authors thank Tarbiat Modares University (TMU) and Iran National Science Foundation (INSF) for supporting this study.

## Author Contributions

**Conceptualization:** Hossein Abolhassani, Mohammad Zaer, Seyed Abbas Shojaosadati, Sameereh Hashemi-Najafabadi.

**Data curation:** Hossein Abolhassani, Mohammad Zaer, Seyed Abbas Shojaosadati.

**Formal analysis:** Hossein Abolhassani, Mohammad Zaer.

**Investigation:** Hossein Abolhassani, Mohammad Zaer.

**Methodology:** Hossein Abolhassani, Mohammad Zaer, Seyed Abbas Shojaosadati, Sameereh Hashemi-Najafabadi.

**Project administration:** Seyed Abbas Shojaosadati.

**Resources:** Seyed Abbas Shojaosadati, Sameereh Hashemi-Najafabadi.

**Software:** Seyed Abbas Shojaosadati.

**Supervision:** Seyed Abbas Shojaosadati, Sameereh Hashemi-Najafabadi.

**Validation:** Hossein Abolhassani, Seyed Abbas Shojaosadati, Sameereh Hashemi-Najafabadi.

**Writing – original draft:** Hossein Abolhassani.

**Writing – review & editing:** Seyed Abbas Shojaosadati.

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
