## [Decision Letter · Decision Letter 0]

24 Nov 2022

PONE-D-22-26320Rapid Generation of Homogenous Tumor Spheroid Microtissues in a Scaffold-Free Platform for High-Throughput Screening of a Novel Combination NanomedicinePLOS ONE

Dear Dr. Seyed Abbas Shojaosadati,

Thank you for submitting your manuscript to PLOS ONE. After careful consideration, we feel that it has merit but does not fully meet PLOS ONE’s publication criteria as it currently stands. Therefore, we invite you to submit a revised version of the manuscript that addresses the points raised during the review process.

We look forward to receiving your revised manuscript.

Kind regards,

Gantumur Battogtokh

Academic Editor

PLOS ONE

Journal Requirements:

"This study was financially supported by Tarbiat Modares University (TMU) and Iran National Science Foundation (INSF)."

Reviewers' comments:

Reviewer's Responses to Questions

**Comments to the Author**

1. Is the manuscript technically sound, and do the data support the conclusions?

Reviewer #1: Yes

Reviewer #2: Partly

Reviewer #3: Yes

2. Has the statistical analysis been performed appropriately and rigorously? 

Reviewer #1: Yes

Reviewer #2: Yes

Reviewer #3: Yes

3. Have the authors made all data underlying the findings in their manuscript fully available?

Reviewer #1: Yes

Reviewer #2: Yes

Reviewer #3: Yes

4. Is the manuscript presented in an intelligible fashion and written in standard English?

Reviewer #1: Yes

Reviewer #2: Yes

Reviewer #3: Yes

5. Review Comments to the Author

Reviewer #1: The authors presented a study on the screening of nanomedicine on tumor spheroids.

They constructed the adhesive-free concave surface and loaded the assembled nanodrugs for testing tumor inhibition efficiencies at varied component ratios and delivery methods. They addressed on the surface treatment, drug synthesis and drug intake quantitation process and tested the drug efficacy in 3D spheroids. However, issues listed below should be settled before being accepted.

1. Page 16, line 1. Typos in “MDA-MB-23”. The missing number should be implemented.

2. The figures should be combined together to include more information in one figure. And, the quality (font consistency, color matching, boldness, arrows, resolution) should all be improved profoundly. There must be neat ways of exhibition for Figure 3, 4, 12.

3. Some of the figures are supposed to move into supp. Figures.

Reviewer #2: In this work, the authors show how three compounds encapsulated in nanoparticles are used to test drug efficacy in 2D cell cultures and 3D spheroids obtained from a breast cancer cell line.

Overall, the paper lacks novelty and clarity in few aspects of the proposed work. Therefore, I do not recommend publication in its present form. Below are my comments:

1) The introduction is rather lengthy and it would benefit from a clear summarised statement with regard to what aspect(s) of in vitro preclinical assays this work is addressing.

2) It is unclear what the original aspects of this work are. A statement of novelty should be inserted. Is this about the NP formulation? Running NP-based assays in low adhesion plates using spheroids is not novel, so maybe is the results obtained? These points should be clearly addressed to remove ambiguities.

3) The authors state the importance of using physiologically relevant models of cancer to bridge the gap with in vivo models, but here a single cell line is used which is far from any TME complex composition. So one may question the suitability of the model used (for example due to the lack of ECM, fibroblasts or other cell types in an avascular tumour tissue).

4) The bottom image in the table on page 10 is blurry and it is unclear what is representing. The table itself is redundant and the points have already being discussed in the introduction.

5) Very importantly, what is the advantage of using the proposed NPs if the drug efficiency was higher for free drugs (i.e. without NPs)? What is the evidence of “enhanced penetration and combinational therapy effect were practically translated into the tumour spheroid cytotoxicity after a 48 h treatment” as stated in section 3.7? I cannot find this in the paper.

6) It is well established that 2D cell monolayer respond differently to 3D cultures, but the model used here is not similar to any in vivo tumours.

7) Regarding the wound healing assays, why was this not done with free drugs as well as a comparison? The conclusion obtained from this assay are not meaningful in my opinion without a comparison with free drugs.

8) Where is the evidence of deep NP penetration? It is stated that NPs could not penetrate tight spheroids! So this is a disadvantage with respect to free drugs.

9) The statement about HTS with plates and models is not justified. The authors present a screen done in low adhesion 96 well plates. What is new about this? What is different from any other HTS assays in plates (e.g. 96 or even higher 384 well plates)?

10) Figure 9 is just an estimate schematic, to show proliferative, quiescent and necrotic zones, but specific dyes should be used to show the different phenotypes in a spheroid.

Reviewer #3: The work has good design and contains valuable information on “Rapid Generation of Homogenous Tumor Spheroid Microtissues in a Scaffold-Free Platform for High-Throughput Screening of a Novel Combination Nanomedicine”. I am sure that the paper will give further insights into new studies related to tumor spheroids for the screening of nanomedicines. However, I suggested the paper acceptance after major corrections.

1. Introduction part of the manuscript is too long. Please re-write and reduce the size of the introduction to almost half.

2. The methodology part is unclear. This seems like the author skipped many important information.

3. Why did the author use PVA coating instead of using low attachment plates for generating 2D and 3D spheroids?

4. There were no clear information about 3D spheroid formation in the same methodology part.

5. Did author use EGF and FGF growth factors in the growth media of tumor spheroids??

6. The number of cells used for spheroids formation is too high for 96 well plates. The author should provide the proper justification with citations for using that much high number of cells.

7. Cell density used per well for scratch wound healing assay is also very high as author used 24 wells plate. Please provide references considering during the experiment performed.

8. In many places the author cited the mechanism such as drug induces tumor cell apoptosis via various apoptosis signaling pathways such as NF-kB, PI3K/Akt, and STAT3. This is totally irrelevant in the whole manuscript as the author did not do anything related to mechanism throughout his study.

9. Figure 2B: I would like to see the complete crop image of Particle Size Distribution Graph mentioning the size of particles.

10. I do not find the Zeta Potential Graph of particle. I would recommend please including the Zeta potential graph as well as PDI values.

11. There are no scale bars in figure 10, figure 11, and figure 13. Please provide the scale bars for all these images.

12. I suggested language checking of the manuscript before publication because some mistakes are looked in the manuscript.

13. Please include some important references during revising the manuscript suggested below:

https://doi.org/10.1016/j.fct.2019.01.018

https://doi.org/10.1016/j.mtchem.2021.100618

https://doi.org/10.1016/j.msec.2019.110285

6. PLOS authors have the option to publish the peer review history of their article (what does this mean?). If published, this will include your full peer review and any attached files.

Reviewer #1: No

Reviewer #2: **Yes: **Michele Zagnoni

Reviewer #3: No

---

## [Author Response · Author response to Decision Letter 0]

5 Jan 2023

The questions and comments received from the editor and reviewers are answered and addressed meticulously in the attached file entitled "Response to Reviewers".

---

## [Editor Report · Decision Letter 1]

7 Feb 2023

Rapid generation of homogenous tumor spheroid microtissues in a scaffold-free platform for high-throughput screening of a novel combination nanomedicine

PONE-D-22-26320R1

Dear Dr. Seyed Abbas Shojaosadati,

We’re pleased to inform you that your manuscript has been judged scientifically suitable for publication and will be formally accepted for publication once it meets all outstanding technical requirements.

Kind regards,

Gantumur Battogtokh

Academic Editor

PLOS ONE
---

## [Editor Report · Acceptance letter]

9 Feb 2023

PONE-D-22-26320R1 

Rapid generation of homogenous tumor spheroid microtissues in a scaffold-free platform for high-throughput screening of a novel combination nanomedicine 

Dear Dr. Shojaosadati:

I'm pleased to inform you that your manuscript has been deemed suitable for publication in PLOS ONE. Congratulations! Your manuscript is now with our production department. 

Kind regards, 

on behalf of

Dr. Gantumur Battogtokh 

Academic Editor

PLOS ONE